

# Peat decomposability in managed organic soils in relation to land-use, organic matter composition and temperature

Cédric Bader[1,2], Moritz Müller[3], Rainer Schulin[2], Jens Leifeld[1]

[1] Climate / Air Pollution Group, ISS Agroscope, Zurich, 8046, Switzerland
[2] Inst Terr Ecosyst, ETH Zürich, 8092, Switzerland
[3] School of Agricultural, Forest and Food Sciences, Bern Univ of Applied Sciences Zollikofen, 3052, Switzerland

*Correspondence to:* Jens Leifeld (jens.leifeld@agroscope.admin.ch)

**Abstract.** Organic soils comprise a large yet fragile carbon (C) store in the global C cycle. Drainage, necessary for agriculture and forestry, triggers rapid decomposition of soil organic matter (SOM), typically increasing in the order forest < grassland < cropland. But there is also large variation in decomposition due to differences in hydrological conditions, climate, and specific management. Here we studied the role of SOM composition on peat decomposability in a variety of differently managed drained organic soils. We collected a total of 560 samples from 21 organic cropland, grassland and forest soils in Switzerland, monitored their $CO_2$ emission rates in lab incubation experiments over 6 months at two temperatures (10 and 20°C) and related them to various soil characteristics, including bulk density, pH, soil organic carbon (SOC) content, and elemental ratios (C/N, H/C and O/C). $CO_2$ release ranged from 6 to 195 mg $CO_2$-C g$^{-1}$ SOC at 10°C and from 12 to 423 mg g$^{-1}$ at 20°C. This variation occurring under controlled conditions suggests, that besides soil water regime, weather and management, SOM composition may be an underestimated factor that determines $CO_2$ fluxes measured in field experiments. However, correlations between the investigated chemical SOM characteristics and $CO_2$ emissions were weak. The latter also did not show a dependence on land-use type, although SOM characteristics revealed that peat under forest was decomposed the least. High $CO_2$ emissions in some topsoils were probably related to the accrual of labile crop residues. Temperature sensitivity of decomposition (Q10 on average 2.57 ± 0.05) was the same for all land-uses, but lowest in soil depths below 60 cm in croplands and grasslands. This, in turn, indicates a relative accumulation of recalcitrant peat in topsoils. Hence, our data suggest that after exposure of subsoil peat in the future, carbon loss from agriculturally managed organic soils will be similar considering warmer climate conditions.

## 1. Introduction

Organic soils represent a major global sink for atmospheric carbon (C). Although they cover only 3 % of the earth's terrestrial surface (Joosten, 2004), they store up to 30 % of the global soil organic carbon (SOC) pool (Parish et al., 2008). In Europe, more than 50 % of the former peatland area, containing organic soils, has been destroyed by peat mining and conversion of land use, including drainage to improve their suitability for agriculture or forestry (Joosten, 2009). Drainage has the purpose to aerate the soil so that plants of interest for agriculture and forestry can grow and make these soils manageable. The change from anaerobic to aerobic





conditions, however, triggers rapid decomposition of peat that had accumulated under the conditions of waterlogging. This transforms the former C-sink into a major source of atmospheric carbon dioxide ($CO_2$) and makes peatlands an important contributor to global climate change (Freeman et al., 2004). Around 85% of the

global annual $CO_2$ emission of 915 Mt $CO_2$-C from drained peatlands are estimated to originate from organic soils now used as croplands (Tubiello et al., 2016). With rates of 6.5 - 9.4 t C ha$^{-1}$ a$^{-1}$ net $CO_2$ fluxes from organic soils now used as croplands were in average found to be higher than from organic soils under grassland, which were estimated to vary between 1.8 and 7.3 t C ha$^{-1}$ a$^{-1}$ (IPCC, 2014). However, recent studies reported emission rates of 7.6 ± 2.0 t C ha$^{-1}$ a$^{-1}$ on organic soils managed as grassland in Germany and thus much higher

rates than previously found for this type of land-use (Tiemeyer et al., 2016). Drained organic soils under forest can act as both, net sinks or sources of atmospheric $CO_2$ (Cannell et al., 1993; Minkkinen & Laine, 1998; Minkkinen et al., 1999; Wüst-Galley et al., 2016), although they are in general considered to represent a source with average net $CO_2$ emissions of 2.0 – 3.3 t C ha$^{-1}$ a$^{-1}$ in the temperate zone (IPCC, 2014). Temperature and soil moisture regime, which depends among others on drainage depth, are main factors influencing peat decay in

drained organic soils (Hogg et al., 1992; Berglund, 1995; Scanlon & Moore, 2000; Chimner & Cooper, 2003; Couwenberg et al., 2010; Leifeld et al., 2012). However, there are substantial differences in $CO_2$ emissions from organic soils with similar drainage and cultivation properties. Protection of organic matter (OM) against decomposition by mechanisms such as occlusion in aggregates and binding to mineral surfaces, which are important for the stabilization of OM in mineral soils (Six et al., 2002), are of minor importance in organic soils

due to the lack or low abundance of minerals (Han et al., 2016). Therefore, the intrinsic decomposability of organic matter is considered another major factor influencing the rate of peat decomposition and a major reason of substantial variation in $CO_2$ emissions among different sites (Chimner & Cooper, 2003; Byrne & Farrell, 2005; Höper, 2007; Wickland & Neff, 2008; Reiche et al., 2010).

Although intrinsic decomposability of (SOM) cannot be addressed directly, useful indicators of the latter are the

relative abundances of labile and recalcitrant C moieties, which shift towards progressively higher proportions of the recalcitrant C with decomposition (Beer et al., 2008; Tfaily et al., 2014). It is important to recognize that during peat formation, most of the net primary production contained in the initial mass of plant residues are lost due to mineralization, and only 10-20% is transformed and accumulated as peat in the water saturated zone of a peat bog or fen (Clymo, 1984). Although decomposition acts slowly on accumulating peat of undisturbed, i.e.

water saturated organic soils, it is believed that primarily the most labile OM moieties are lost. Due to fresh peat layers accumulating on top of older ones, age and depletion in labile compounds increase with soil depth. Incubation studies of peat samples and carbon loss studies with undisturbed organic soils found smaller $CO_2$





emission rates from deeper peat layers, which was related to the absence of labile compounds i.e. a lower

intrinsic decomposability of soil organic matter (SOM) (Hogg et al., 1992; Scanlon & Moore, 2000; Wang et al.,

2010; Hardie et al., 2011; Leifeld et al., 2012). Using solid-state [13]C-NMR, DRIFT/FTIR spectroscopy, and

pyrolysis-GC/MS, various studies of OM composition of undisturbed peat profiles have shown a gradual change

with increasing depth towards a relative enrichment of compounds that are recalcitrant against decomposition

under anoxic conditions, such as lignins and polyphenols (Freeman et al., 2004), while the contents of labile

oxygen-rich compounds, such as polysaccharides, were found to decrease (Leifeld et al., 2012; Biester et al.,

2014; Sjögersten et al., 2016).

Elemental ratios between oxygen (O), hydrogen (H), nitrogen (N) and carbon are widely used as indicators of the

relative abundance of different groups of compounds such as phenols, lipids and polysaccharides, and proteins.

Lignins and polyphenols have molar O/C ratios in the range of 0.2 – 0.6 and H/C ratios between 0.9 and 1.5,

while the respective ratios of carbohydrates range from 0.8 to 0.9 for O/C and from 1.4 to 1.8 for H/C (Kim et

al., 2003). In line with the molecular and spectroscopic analyses mentioned before, both ratios were found to

decrease with increasing depth in peat (Klavins et al., 2008; Biester et al., 2014; Wüst-Galley et al., 2016). On

the other hand, both, fresh plant residues and undisturbed peat usually have high C/N ratios (Loisel et al., 2012).

When peat becomes exposed to oxic conditions, mineralization seems to lead to relative enrichment of N,

explaining why decreased C/N ratios are found in organic topsoils compared to undrained peat layers or bottom

layers of drained organic soils (Malmer & Holm, 1984; Kuhry & Vitt, 1996; Krueger et al., 2015). While

undisturbed organic soils have a low bulk density, drainage leads to subsidence processes and increasing bulk

densities in the topsoils (Rogiers et al., 2008; Leifeld et al., 2011a; b).

The temperature sensitivity of peat mineralization, as expressed by its Q10 value, is a useful parameter to

characterize the intrinsic decomposability of SOM (Hogg et al., 1992; Biasi et al., 2005; Davidson & Janssens,

2006; Conant et al., 2008; Boddy et al., 2008; Karhu et al., 2010; Hilasvuori et al., 2013). In line with the

biochemical and elemental evidence reviewed above, it was reported to increase with increasing resistance of

peat soils against OM decomposition (Scanlon & Moore, 2000), soil depth and peat age (Hardie et al., 2011;

Hilasvuori et al., 2013).

Despite its likely important role in determining future C losses from drained peatland, the influence of SOM

composition on peat decomposition in managed organic soils is not well studied. While decomposition rates

seem to decline with increasing peat age i.e. profile depth, the oxic conditions, occurring after drainage onset,

lead to fast SOM decomposition. As for undisturbed organic soils, we expect that post drainage decomposition

primarily acts on the most labile OM moieties. However, the much faster decomposition of labile SOM might

alter the depth interaction found in undisturbed peat soils. Further, recent inputs from plant residues may supply

the topsoils with labile OM. The fraction of crop residue derived and thus more rapidly decomposing SOM might account for at least 20% of carbon in agriculturally managed organic soil (Bader et al., 2017). The fractions of OM derived from peat and recent inputs and their decomposability in drained organic soils may, however, substantially vary with land-use, site conditions and time since land use conversion. (Schulze et al., 2009) reported that inputs of fresh organic matter residues were smaller in croplands than in grasslands or

forests, suggesting that SOM might be on average more aged and thus less decomposable. In-situ measurements of $CO_2$ fluxes from managed organic soils reveal slower decomposition of peat under forest (IPCC, 2014). Together, smaller peat loss rates and higher residue input make us expect that SOM decomposition rates under controlled conditions are fastest in forest topsoils.

In this study, we analysed the relationship between SOM properties, specific decomposition rates ($CO_2$-C mg$^{-1}$

SOC) and their temperature sensitivity of peat samples taken from depths between 0 and 200 cm of 21 drained organic soils in Switzerland managed as cropland, perennial grassland or forest. We measured decomposition rates in incubation experiments under standardized lab conditions and interpreted the current decomposition status of peat using SOM properties such as i) carbon stocks, bulk densities and the elemental ratios O/C, H/C and C/N as well as ii) the temperature sensitivity towards decomposition, expecting that

1. Specific decomposition rates of SOM decline with depth

    2. Specific decomposition rates of SOM in managed peatlands correlate with its composition and are inversely related to the temperature sensitivity of decomposition

    3. Specific decomposition rates of topsoil SOM are largest in the forest and smallest in the croplands.

## 2. Methods

### 2.1 Sampling site

The soil samples used for this study were taken from organic soils distributed across Switzerland. Apart from current land-use (grassland, cropland, forest), they differed in the type of drainage system (ditches in forest, pipes in crop- and grassland), time since drainage onset, altitude (MASL), and climate (Tab. 1). All sites were classified as fens, although we found bog-derived peat layers within the top 30 and 40 cm of the soil profiles at

two sites (SKF, KF).

### 2.2 Soil sampling

Between October 2013 and June 2015, we sampled in total 88 peat cores from all 21 sites (4 cores per site whereas 1 site had 8 cores). While the 4 cores from one particular site were between 1 and 2 m long, all cores

from the other sites had a maximum length of 1 m. If the mineral layer was reached before 1 m depth, coring was

discontinued. We used a Belorussian peat corer for soils with low bulk densities and a motorized Humax corer

for denser soils. The samples were stored at 4 °C for up to 2 months until analysis. We applied the method of

(Rogiers et al., 2008) to account for soil compaction during sampling, and divided the cores into segments

corresponding to 5 - 10 cm depth increments. In total this resulted in 1605 soil samples. Some cores had

interlayers of mineral sediment. These interlayers were excluded from the analysis. The soil of one site (BIF) had

no limnic layer and therefore was classified as a murshic histosol; all others were classified as murshic limnic

histosols (WRB, 2014).

### 2.3 Soil analysis

Soil pH was measured for two to three samples of each core (307 samples in total), using a flat surface electrode

(pH 100, Extech Instruments, USA) calibrated at pH 7.00 and pH 4.01. Aliquots of fresh soil (10 g dry matter)

were diluted in distilled water (2.5 parts water to 1 part material by mass), shaken, left for 20 hours and shaken

again, before the pH measurements were carried out.

Prior to further chemical analysis, the samples were oven-dried at 105 °C and weighed to determine bulk density

($g\ cm^{-3}$). The dried samples were ground for 2 min at 25 rotations $s^{-1}$ in a ball mill (Retsch MM400) and

subsampled to determine total carbon (Ctot), SOC, hydrogen (H), nitrogen (N) and oxygen (O) contents. Ctot, H

and N were analysed after dry combustion of ground subsamples in an elemental analyser (Hekatech, Germany).

To determine SOC, we hydrolysed ground aliquots with 36 % HCl (acid fumigation) in a desiccator to remove

any carbonates, before the samples were analysed in the elemental analyser. A third set of ground subsamples

were used to determine the O contents by means of the same analyzer after pyrolysis at 1000°C. We corrected O

contents for inorganic O, assuming that all inorganic O was present in form of $CaCO_3$. The O/C and H/C ratios

given in this paper represent mole ratios, whereas the C/N ratios represent mass ratios. Soil carbon stocks (t C ha$^{-1}$) refer to the organic horizons summed over each profile and thus do not include sediment layers that

interspersed the profiles.

### 2.4 Incubation experiment

We selected at least two soil segments of each soil core from depths between 0-30 cm, 30-60 cm and 60-100 cm

for incubation to determine SOM decomposability. Half of the samples were incubated for at least six months at

10 °C and the other half at 20 °C. From the one location (MCL) where we had taken cores of >100 cm length,

we selected six additional samples from the depth below 100 cm for incubation, resulting in a total of 560

incubated samples. Prior to incubation, we thoroughly mixed every segment, removed visible roots and adjusted

the water potential to -10 kPa, using a hanging water column. The sample weight was 53.9 ± 0.7 g (mean ±





standard error) at -10 kPa. Following the method of (Chapman, 1971), we measured $CO_2$ emission rates by means of a Respicond VII analyser (Nordgren Inovations, Sweden) over three to four measurement cycles of several weeks between November 2013 and March 2016. The measurement principle is based on the change in electrical conductivity of the NaOH solution with increasing uptake of $CO_2$. In each cycle, we vented the alkali $CO_2$ traps (NaOH 0.6 M) of the analyser regularly after 50-60 mg of $CO_2$ had been emitted to prevent $O_2$

deficiency. In addition, we exchanged the NaOH solution while the traps were vented. Between measurement cycles, we kept the soil samples at the same temperature and moisture level as during the cycles.

### 2.5 Data analysis

We only used $CO_2$ data taken after the first 3 days of each measurement cycle for data analysis to avoid artefacts that might have resulted from moving the samples and adjusting their water content. Furthermore, we excluded

all negative emission rate values (0.45 % out of 1700 $CO_2$ measurements taken on average per sample). Data gaps (83 % of the timeline) between measurement cycles were filled by means of interpolation using a robust linear regression on the log transformed data. The specific amount of SOC which was lost from a sample as $CO_2$ during 10,000 hours of incubation at 10 or 20 °C [mg $CO_2$-C $g^{-1}$ SOC], L, was calculated as

$$L = \frac{(CO_{2\,sample} - CO_{2\,blank}) \times \frac{12.01}{44.01}}{SOC_{sample} \times m_{sample}} \qquad (1)$$

where $CO_{2\,sample}$ is the amount of $CO_2$ lost from the sample over 10,000 hours of incubation [mg $CO_2$-C $g^{-1}$ SOC], $CO_{2\,blank}$ is the median of ambient $CO_2$ accumulation collected in 6 blank vessels over more than 6 months and extrapolated to 10,000 hours (on average 27 mg), $SOC_{sample}$ is the SOC content of the sample [g $kg^{-1}$], and $m_{sample}$ is the mass of the soil sample [kg].

To determine Q10 values we used the method used by e.g (Hogg et al., 1992; Scanlon & Moore, 2000; Wang et

al., 2010; Wetterstedt et al., 2010; Hardie et al., 2011), dividing the 10,000 h length of the incubation period at 10 °C by the time span over which samples incubated at 20 °C emitted the same amount of $CO_2$-C per mg SOC as those incubated at 10 °C emitted during 10,000 h. Given that the same amount of SOC is lost at both temperatures, changes in OM composition during incubation are also assumed to be the same and thus differences in the rates are assumed to reflect only the influence of temperature and not that of differences in

composition. Q10 values are known to depend on incubation temperatures. In order to compare our results with those of other studies we calculated the activation energy (Ea in kJ $mol^{-1}$) required for decomposition of SOC, using Q10 values:

While R is the gas constant (8.314 J $K^{-1}$ $mol^{-1}$) and T is the temperature used for incubation (K).

$$EA = \frac{R \times \frac{\ln(Q10)}{(\frac{1}{T1} - \frac{1}{T2})}}{1000} \qquad (1)$$





Mixed linear models were used to analyse the effects of the various soil parameters on SOM mineralization and

their interactions with land-use. The function lmer from the package lme4 (Bates et al., 2015) was implemented

using the software R (R core Team, 2015) to run mixed linear models. Heteroscedasticity or departure from

normality was assessed graphically. In order to avoid heteroscedasticity, we log-transformed topsoil C-stocks

and bulk density data. We tested whether the factor "land-use" had a significant influence on the variation of

each of the analysed variables ($\alpha = 0.05$). To do this, the following two mixed models, [2] and [3], were run for

each dependent variable and compared using an ANOVA.

*variable~land.use + random effects*                    *(2)*

*variable~random effects*                    (3)

Sampling depth, sampling location and site repetition were included as random effects to account for the

dependence among segments of the same core and among cores from the same sampling location, respectively.

Further, we determined the significance of land-use specific differences (CL vs FL, CL vs GL, FL vs GL), using

a least square means test for linear models (lsmeans package).

We used the same approach to test the influence of the factor 'soil depth' on the target variables with interactions

between the three sampling depths (0-30 cm, 30-60 cm, >60cm), using the model

*variable~depth.interval + random effects*                    *(4)*

in addition to Equation 3. To determine the significance of depth-specific differences, we used least square

means test as mentioned before.




## 3. Results

### 3.1 SOM characteristics

Soil pH, SOC content, C/N ratio and bulk density showed significant land-use effects (Fig. 1; Tab. 2; Tab. S1): The lowest soil pH value were found in the forest topsoil samples, whereas SOC content and C/N ratio were the highest in these samples. Bulk density was highest in the cropland topsoils. Below 30 cm depth, soil pH, SOC content, C/N ratio and bulk density showed no land-use effect.

In the forest soil profiles, soil pH overall increased with depth, whereas it decreased in the grassland and

215 cropland soils (Tab. 3). Also bulk density decreased with depth in the grassland and cropland soils, while SOC content and C/N ratio increased. In the forest soils, SOC content, bulk density and C/N ratio did not differ between topsoil (0-30 cm depth) and subsoil samples (30-60 cm depth); however, below 60 cm depth SOC was slightly lower than above, while bulk density and C/N ratio were higher than above 60 cm depth (Fig. 1, Tab. 3). The cumulated topsoil C stocks showed no land-use effects, but tended to be larger in cropland and forest than in

grassland soils over the entire profile (Fig. 1; Tab. 2).

The molar H/C and O/C ratios of the organic matter fell between the typical values of the ratios for carbohydrates and lignin, which is displayed in a Van Krevelen plot (Fig. 2). The lowest values of both ratios were found in the forest soils, the highest in the grassland and cropland topsoils. Both ratios were lower in the topsoils than in the subsoils of the cropland and grassland sites, while there was no difference between the two

depths in the forest soils (Tab. 3). At depths below 30 cm, the O/C ratio was lower in the forest soils than in the other soils, but without land-use effect in the H/C ratio.

### 3.2 $CO_2$ emissions and Q10

The samples incubated at 10 °C emitted $32.56 \pm 1.39$ mg $CO_2$-C g$^{-1}$ SOC, while samples incubated at 20 °C lost

$74.06 \pm 2.98$ mg $CO_2$-C g$^{-1}$ SOC (Fig. 1). At 10°C we did not observe a land-use effect on $CO_2$ emission (Fig. 1, Tab. 2), but at 20 °C the topsoil samples from croplands emitted less $CO_2$ than those from forests. This effect occurred due to extra ordinarily high emissions of the samples from two grassland and two forest sites (VWGL, VWF, SKGL, SKF) (Fig. 3). Those four sites experienced the least intensive drainage. Furthermore, these sampling sites were situated at high altitude in a pre-alpine environment with lower mean annual temperatures

and higher precipitation than at the other sites (Tab. 1). In pairwise comparisons between adjacent sites of different land-use (i.e., VWGL vs VWF, SKGL vs SKF, CM vs CGL and CLG vs GLG and FG (Tab. 1)), land-use effects were only found for the last site (Fig. 3).



At 10 °C, $CO_2$ emissions of the topsoil samples from all sites together were higher than from samples taken at 30

to 60 cm depth, independent of land-use (Tab. 3). Analysing the influence of depth separately by land-use type,

this effect was only manifested in grassland and forest, but not in cropland soils. We found no overall depth

effect at 20 °C, but $CO_2$ emissions of topsoil samples from forests were higher than those of samples taken at

lower depths, whereas we found the opposite for the cropland soils. Despite the just mentioned depth effects, the

general relationship between emissions and soil depth was weak and not consistent in its sign (Tab. 4).

Over the course of the incubation, $CO_2$ emissions increased for 40 % of the samples, as revealed in Tab. S1 by

positive slopes of the regression lines. These increases were independent of land-use. In total, the $CO_2$ emissions

from these samples were almost 50 % higher than those from the other samples that instead showed a trend of

decreasing emissions.

Mean Q10 values were 2.57 ± 0.05. The Q10 did not differ between the three land use types. It was lower below

60 cm depth in the cropland and grassland, but not in the forest soils (Fig. 1; Tab. 3). Activation energies (Ea)

calculated from Q10 values ranged around 48.1 and 123.5 kJ mol$^{-1}$ and like Q10 values decreased with depth.

There were significant relationships between $CO_2$ emission and SOC content, bulk density and C/N ratio in

general but they were weak (Tab. 4). The Q10 values showed similar relationships to these soil variables as $CO_2$

emission.

## 4. Discussion

### 4.1 SOM characteristics

The SOC contents, bulk densities and C/N ratios found in the deeper parts of our soil profiles were close to

values that are typical for undisturbed peat (Grover & Baldock, 2012; Loisel et al., 2014). They also indicate that

our soils were characteristic for European fens and resembled typical properties of managed organic soils

(Berglund, 1995; Kechavarzi et al., 2010; Eickenscheidt et al., 2015; Krueger et al., 2015; Wüst-Galley et al.,

2016; Brouns et al., 2016). Several studies assume that bottom layer peat of managed organic soils is less

decomposed (Ewing & Vepraskas, 2006; Rogiers et al., 2008; Leifeld et al., 2011b; b; Krueger et al., 2015;

Wüst-Galley et al., 2016). We therefore interpret the different SOM characteristics found in the topsoils of our

samples as indicators of an advanced decomposition triggered by drainage.

The land-use specific differences manifested in different topsoil SOC contents and C/N ratios (highest under

forest), and topsoil bulk densities (lowest under forest). The higher forest C/N ratios might be explained by

absence of the use of N fertilizers and lower bulk densities by lower traffic with field machinery. In addition,





differences in C/N between land-use types may also suggest that peat decomposition was less advanced in
forests compared to croplands and grasslands. Further, depth effects are lowest in forest soils, indicating a lower

impact of soil management that could also result in a lower decomposition of forest topsoils. The relatively high
carbon stocks found in cropland top soils are most likely the result of subsidence after drainage and compaction
from field traffic, leading to increased soil bulk density in the uppermost layers. This effect, with respect to C
stocks, overrides the overall much smaller C concentration in agriculturally managed organic soils.

The H/C and O/C ratios in the bottom layers of our soils were similar to those found in undisturbed bogs and

drained bogs used for forestry in Switzerland (Zaccone et al., 2007; Wüst-Galley et al., 2016). They indicate an
enrichment of polyphenols with depth, which is in line with the current understanding of peat development in
peatlands (Cocozza et al., 2003; Zaccone et al., 2007; Klavins et al., 2008; Delarue et al., 2011; Leifeld et al.,
2012). The increased H/C and O/C ratios in the grass- and cropland topsoils can be attributed to inputs of fresh
plant litter to the topsoil via above- and belowground residues, as such residues are rich in carbohydrates

(Koegel-Knabner, 2002). In a previous study, in which we used stable and radiocarbon isotopes to label the SOC
of two of our soils (CM and CGL in Table 1), at least 20 % of topsoil organic matter was no peat, but derived
from recent plant litter (Bader et al., 2017). The results further indicated that the OM derived from these fresh
plant residues were a source of labile C that contributed more to decomposition than the old, peat-derived SOM.
The H/C and O/C ratios reflect the mixing ratio of these two SOM sources. The H/C and O/C ratios in forest

topsoils were lower than of those under cropland and grassland and did not change with depth. Interpreting these
lower H/C and O/C ratios in the forest top-soils as indicators of more advanced peat decomposition (Klavins et
al., 2008; Leifeld et al., 2012; Biester et al., 2014; Wüst-Galley et al., 2016) would be in contradiction to our
conjecture that land management effects on peat decomposition, revealed by SOC, bulk density and C/N ratio,
are less pronounced for forests. We rather argue that reason for the low H/C and O/C ratio in the forest soils is a

higher abundance of lignin rich (wood derived) plant residues. A second mechanism for comparably higher O/C
and H/C ratios in cropland and grassland soils, could be that peat loss in the uppermost layers was higher under
agriculture than under forest, resulting in a relatively higher share of SOM from recent inputs. Considering all
the available evidence SOM characteristics, we conclude that peat decomposition is less advanced in forest soils
than in agricultural soils, in line also with field flux measurements on managed organic soils that typically show

faster decomposition in croplands and grasslands than in forests (IPCC, 2014).



### 4.2  CO₂ emissions and temperature sensitivity of decomposition

Our soils lost, on average, ca. 5-10 % of their SOC during incubation of > one year (20 °C). The advanced decomposition state of many of our samples might give reason to expect that $CO_2$ rates are below that of more

intact peat or mineral topsoils that contain a larger fraction of recent plant residues. To understand whether SOM in our organic soils is particularly stable, we compared its daily carbon loss with data from studies that used undisturbed to extensively managed organic soils, or mineral soils (Tab. 5; Fig. 4). Indeed, our values are on average below those from other organic soil studies. However, their range overlaps with the uncertainty of the regression line that is plotted through results from studies from other, mostly intact or little degraded organic

soils. Hence, the pronounced oxidative decomposition after long periods of drainage might result in a relatively smaller labile SOC pool, but the large variability between experimental set-ups, incubation lengths, and water contents among incubation studies prevents a stronger line of interpretation. Interestingly, the regression lines modelled for organic and mineral soils did not deviate significantly from each other. Therefore, the pools size of labile carbon, indicated by the decomposition rates, seem not to differ between these soil classes. This

comparison suggests that accumulation of recent, labile plant materials that presumably account for most of the evolved $CO_2$ is not systematically different between mineral and organic soils.

Samples showing an increase in $CO_2$ emission rate over time were predominantly of subsoil origin, where SOC contents and C/N ratios indicate a lower decomposition than in the topsoil. Furthermore, based on the information we have on land-use and drainage depths, it appears that most of these samples were taken from soil

layers that were protected from intensive decomposition by high water saturation. The long incubation time in our study might have given aerobic decomposer communities time to develop and grow, which may not be sufficient in shorter studies.

Like other studies on extensively managed or undisturbed organic soils, investigating depth interaction of decomposition rates in the top 30 to 50 cm (Hogg et al., 1992; Scanlon & Moore, 2000; Wang et al., 2010;

Hardie et al., 2011), we found a decrease with depth. However, the relationship between $CO_2$ emissions and depth was rather weak in our case and not consistent for both incubation temperatures and the different land-uses. Compared to the studies on unmanaged organic soils, reporting declines of a factor 2 to 30, our differences were substantially smaller (Hogg et al., 1992; Scanlon & Moore, 2000; Wang et al., 2010; Hardie et al., 2011). Drainage and decadal agricultural use of our soils led to more intense decomposition processes in the topsoil,

resulting in little depth interaction or, for croplands, sometimes maybe even a reversal of decomposability. Further, the abundance and decomposability of crop residues has to be considered as a substantial $CO_2$ source. For two topsoils (CGL and CM), Bader et al. (2017) showed that at least 20% of the SOM is crop residue



derived and responsible for 40% of the emitted $CO_2$. Assuming that the abundance of crop and plant residues is highest in topsoils, it might be possible that decomposability of peat derived SOM either does not depend on depth or topsoil peat decomposes at smaller rates. Therefore, we cannot confirm our first hypothesis of decreasing decomposition rates with depth.

As Tab. 5 shows, the Q10 values found in our study ($2.74 \pm 0.06$) were higher than Q10 values found elsewhere for similar sampling depths but in unmanaged organic soils (Hogg et al., 1992; Chapman and Thurlow, 1998; Scanlon and Moore, 2000; Yavitt et al., 2000; Hardie et al., 2011; Hamdi et al., 2013; Hilasvuori et al., 2013). Also the temperature independent Ea was higher in our samples ($69.4 \pm 3$ kJ mol$^{-1}$) than in most other studies on undisturbed organic soils ($47.4 \pm 7.2$ kJ mol$^{-1}$) (Tab. 5). However, three studies (Chapman, 1971) , (Hardie et al., 2011) and (Hogg et al., 1992) found similar or higher Ea values in northern organic soils. In the case of (Chapman & Thurlow, 1998) they were also managed as grassland or forest, whereas the other studies used peat from undisturbed organic soils. Nevertheless, the high Ea of our samples might reflect the change in chemical peat composition with decomposition after drainage towards higher recalcitrance. In contrast to other studies on unmanaged organic soils reporting no trend or increasing Q10 values with depth (Scanlon & Moore, 2000; Wang et al., 2010; Hardie et al., 2011; Hilasvuori et al., 2013), our cropland and grassland profiles had a lower Q10 below than above 60 cm depth. Various studies on SOM decomposition used Q10 values as an indicator of SOM recalcitrance (Hogg et al., 1992; Biasi et al., 2005; Davidson & Janssens, 2006; Conant et al., 2008, 2011; Hartley & Ineson, 2008; Hilasvuori et al., 2013). Considering that the presence of labile crop residues would decrease Q10 in the topsoil rather than in the subsoil, the higher topsoil Q10 may be explained by an extended accumulation of recalcitrant moieties. This proposed high abundance of recalcitrant moieties in topsoils of degrading organic soils is in line with the reported differences in SOM composition in different layers as well with the pattern of $CO_2$ emissions. The latter show no substantial difference with depth and indicate that a higher fraction of recent and labile plant residues in topsoils is counterbalanced by a high recalcitrance of the highly degraded peat. Comparing radiocarbon concentrations in SOC and emitted $CO_2$ of two sites also used for this study (CM, CGL), Bader et al. (2017) estimated that SOC from plant residue inputs is more labile than peat. The measured radiocarbon contents for SOC were 75 to 80 pMC and therefore indicated that peat of the topsoil must have experienced a substantial decomposition.

It is remarkable that despite the controlled conditions in our incubation experiment the variation in cumulative loss of initial SOC of between 0.6 and 42.3 % (Fig. 4), was similar to or even larger than that observed in field flux measurements (IPCC, 2014). This large variability suggests that the composition of SOM is of similar importance as drainage, climate and other site factors in controlling $CO_2$ emissions from drained organic soils.




Nevertheless, the relationships between the measured SOM parameters to assess the biochemical
decomposability, $CO_2$ emissions and Q10 values were rather weak and thus do not support our second
hypothesis. This stands in contrast to other studies which concluded that chemical composition is a major factor
of SOM decomposability in organic soils (Scanlon & Moore, 2000; Koch et al., 2007; Reiche et al., 2010;
Hardie et al., 2011; Leifeld et al., 2012). However, these studies focused mainly on single profiles of undisturbed
or extensively used organic soils. A recent study investigated relationships between SOM parameters and
decomposition rates of German organic soils under controlled conditions (Säurich et al., 2017). These authors
mostly studied strongly disturbed fens with similar properties to our soils. Besides SOC contents, soil pH and
C/N ratios, Säurich et al., (2017) focused on other soil nutrients, stable isotopes and microbial biomass. In line
with our results, they could not identify strong proxies for SOC decomposition by means of simple chemical
attributes.

In order to explain the weak relationships, it should be considered that, in our case, the emitted $CO_2$ comprised
only 3.2-7.4 % of the total SOC, while the analysed SOM parameters in this and other studies represent bulk
SOM. Our methods allowed to gain a broad overview on the chemical composition of SOM, while
decomposition might more tightly be bound to the abundance of specific OM moieties.

Although land-use affected SOM characteristics, such as elemental contents and their ratios, the amount of $CO_2$
emitted from the soils did not differ among the three types of land-use. We therefore have to reject also our third
hypothesis of a higher SOM decomposition rate in forest topsoils. We assume that long-lasting drainage and
management might have resulted in an equivalent decomposition of most of the labile OM, along with its
intrinsic decomposability.

**5. Conclusions**

Chemical characteristics of SOM indicated an advanced peat decomposition in the uppermost layers of drained
organic soils used as cropland or grasslands. Under controlled moisture and temperature conditions, $CO_2$
emissions from peat samples had a similar variability, as found for in situ $CO_2$ flux experiments on drained
organic soils. Therefore, carbon loss from drained organic soils cannot be explained entirely by climate or
drainage depth. However, simple chemical characteristics of SOM, as used in this study, were not specific
enough to explain the variability in $CO_2$ emissions or the temperature sensitivity of decomposition. Despite that
$CO_2$ emissions were occasionally higher in topsoils, probably derived from accrual of labile plant residues, the
remarkable decrease of Q10 values with depth suggested that the relative content of recalcitrant peat derived



SOM was high in topsoils of managed organic soils, indicating an advanced degradation in these uppermost

layers. It is therefore necessary to quantify the fraction of peat derived SOM throughout a drained soil profile, in

order to verify this assumption. Yet we understand from the similar magnitude of $CO_2$ emission rates found

above and below 30 cm depth that future peat loss will occur at similar or even faster rates, assuming an

increasing mean annual temperature.





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



**Figure 1**

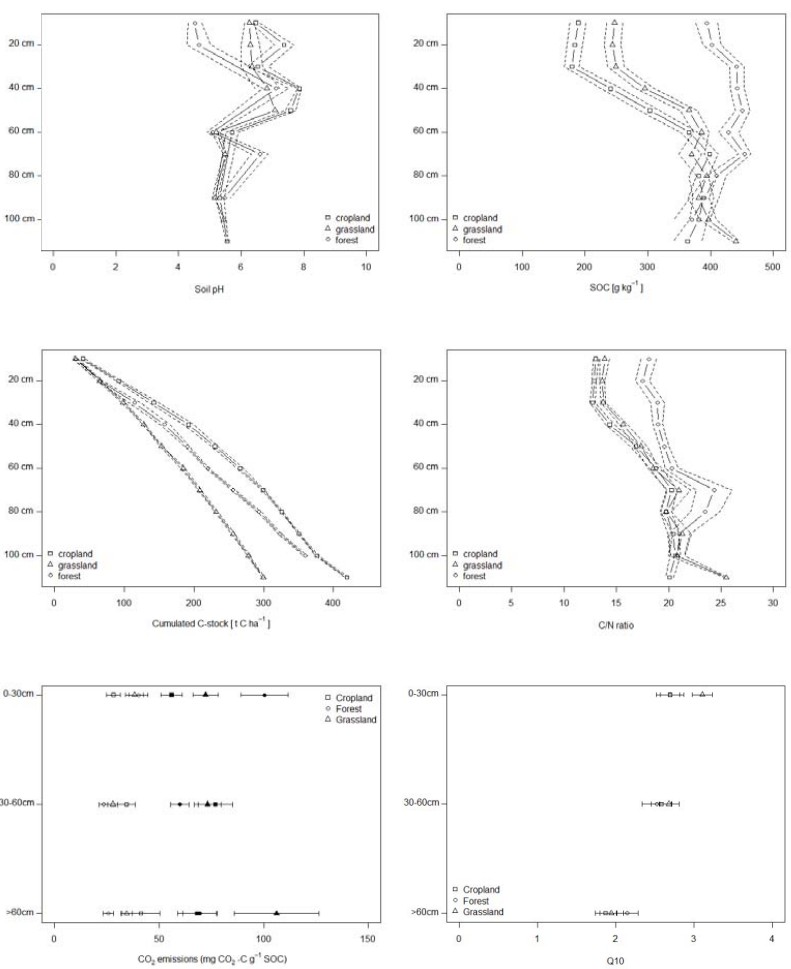

Fig. 1. Soil pH, bulk density, SOC content, cumulated C-stocks, C/N ratios, $CO_2$ emissions and temperature sensitivity (Q10)

displayed for the three land-use types (cropland, grassland and forest). $CO_2$ emissions are displayed at 10 (open symbols) and

20°C (black symbols), while the area between dashed lines and error bars represent the standard errors of the mean.



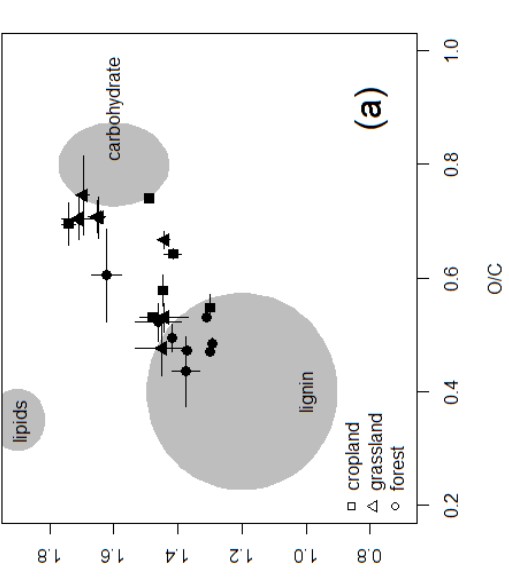

**Figure 2**

Fig. 2. Van Krevelen plots of samples from (a) the upper 30 cm and (b) depths below 30 cm. Symbols represent averages for relevant core segments from each site; black bars represent the standard error of the mean; grey surfaces represent the range of O/C and H/C for lignin, carbohydrates, and lipids, adapted from Preston and Schmidt (2006).





**Figure 3**

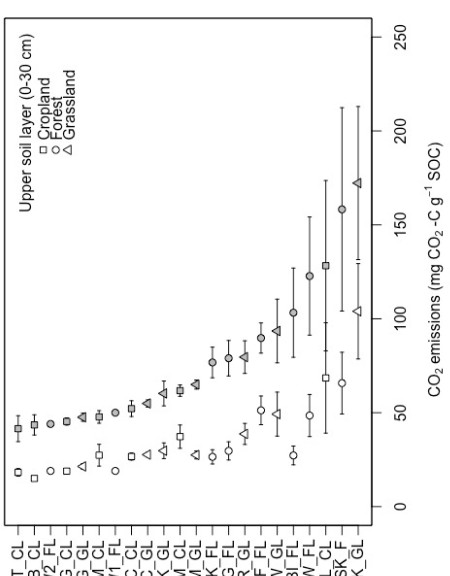

Fig. 3. $CO_2$ emissions at 10 (open symbols) and 20°C (grey symbols) displayed for upper soil layer (0-30 cm) and bottom layers (30-100 cm) of all sampling locations. Error bars represent the standard error of the mean. If a symbol lacks error bars, the standard error was smaller than the symbol size or as in the cases of upper soil layer SW1 and SW2 n= 2.

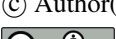



**Figure 4**

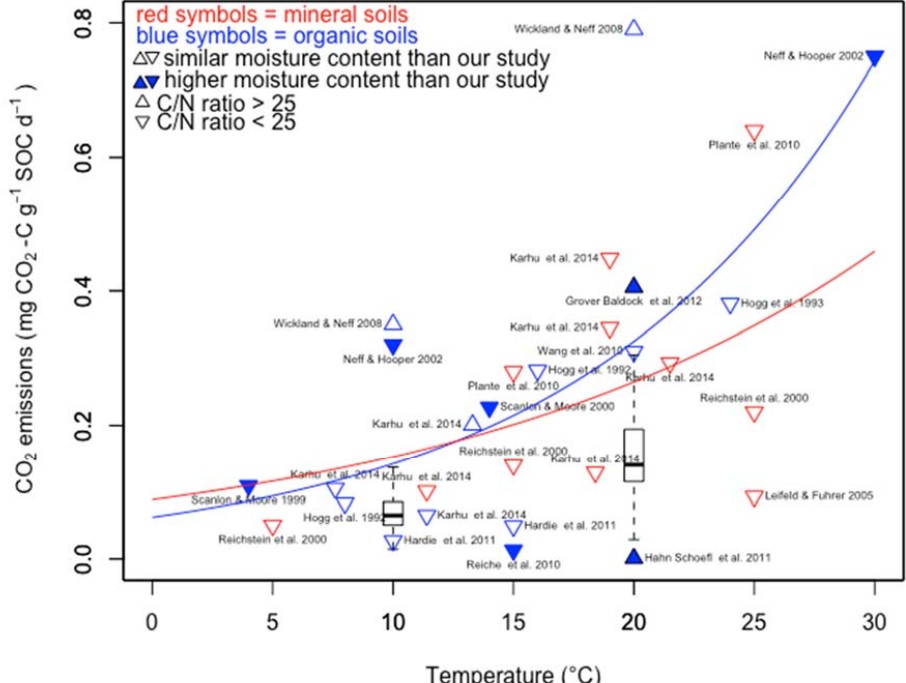

Fig. 4: Comparison of daily $CO_2$ emission rates from this study (boxplots) with rates found during other incubation studies

(organic soils and mineral soils). The curves represent the modelled $CO_2$ emission rates for organic soils from other studies

(blue line) rate=$0.06e^{0.08t}$ and mineral soils (red line) rate=$0.09e^{0.05t}$, for temperatures between 0-30 °C.





**Table 1**

Tab. 1: List of sampling locations, including information on the land-use type, peat thickness, approximate time since drainage onset, elevation (MASL), mean annual temperature (MAT) and mean annual precipitation (MAP) of each site.

| Location name/ Site abbreviation | Land-use (CL: Cropland, GL: Grassland, FL: Forest) | Co-ordinates (WGS 1984) | MASL (m) | Peat thickness[1] (cm) | Drainage history[2] | MAT[3] (°C) | MAP[4] (mm) |
|---|---|---|---|---|---|---|---|
| Gals (G_CL, G_GL, G_FL) | CL, GL, FL | 7.065,47.040 | 430 | <100 | drained by 1864 | 10.0 | 1145 |
| Cressier (C_CL, C_GL) | CL, GL | 7.047,47.041 | 430 | 120 | drained by 1864 | 10.0 | 1145 |
| Brüttelen (B_CL) | CL | 7.175,47.033 | 438 | 290 | drained by 1864 | 9.9 | 1003 |
| Treiten (T_CL) | CL | 7.145,47.010 | 439 | 238 | drained by 1864 | 9.9 | 1033 |
| Staatswald 1+2 (SW1_FL, SW2_FL) | FL | 7.092,46.984 | 431 | 142 | drained by 1864; intensive drainage in 1942 | 10.1 | 990 |
| Mühlethumen (M_CL, M_GL) | CL, GL | 7.523,46.821 7.523,46.817 | 540 | 400 | drained after 1860, intensive drainage in 1942 | 8.9 | 1136 |
| Kirchenthurnen (K_GL) | GL | 7.523,46.821 | 540 | 302 | drained after 1860 | 8.9 | 1136 |
| Birmensdorf (BI_FL) | FL | 8.454,47.357 | 560 | 95 | unclear; peat excavation nearby | 9.2 | 1122 |
| Vorderwengi (VW_GL, VW_FL) | GL, FL | 9.098,47.196 | 1070 | 100 - 146 | grassland drained by 1935 | 6.2 | 2240 |
| Summerigchopf (SK_GL, SK_FL) | GL, FL | 9.399,47.212 | 1300 | 147-202 | drain established between 1935 and 1960 | 6.0 | 1731 |
| Rüthi (R_GL) | GL | 9.536,47.283 | 435 | >700 | drained by 1970 | 10.1 | 1533 |
| Lüchingen (L_CL) | CL | 9.574,47.378 | 414 | 400 | drained by 1860; intensive drainage between 1942 -1962 | 10.1 | 1297 |
| Im Moos (IM_CL) | CL | 9.573,47.379 | 414 | 400 | drained by 1860; intensive drainage between 1942 -1962 | 10.1 | 1297 |
| Katzensee (K_FL) | FL | 8.495,47.433 | 440 | 230 | unclear; peat excavation site nearby until 1940 | 9.4 | 1040 |
| Chreienriet (F_FL) | FL | 8.486,47.434 | 440 | 330 | Unclear; peat excavation site nearby until 1940 | 9.4 | 1040 |

[1]Peat thickness was determined by excavation an additional peat core down to the sediment layer.

[2]Information on drainage was gained by viewing Siegfried topographical maps (1870-1949), considering information on Swiss organic soils by (Lüdi, 1935), as well as aerial photographs. [3]MAT is the average for the years 1981-2010[4]MAP is the average of the years 1971-1991 derived from original data of MeteoSchweiz





**Table 2**

Tab. 2: Results of land-use effect analysis for the whole soil profile as well as specifically in the topsoil (0-30 cm) and bottom layers (>30 cm), displayed for SOC concentration, C-stocks, bulk density C/N, H/C and O/C ratios, $CO_2$ emissions at 10 and 20 °C, and the resulting Q10 values.

| Attribute | Land-use interaction | | P values between spec. land-uses | | |
|---|---|---|---|---|---|
| | $\chi^2$ Value | P value | CL vs FL | CL vs GL | FL vs GL |
| **Soil pH** | $\chi^2(2)=3.7$ | 0.16 | - | | |
| Soil pH (0-30 cm) | $\chi^2(2)=14.9$ | 0.0006 | 0.0003 | 0.9 | 0.0021 |
| Soil pH (>30 cm) | $\chi^2(2)=0.7$ | 0.7 | - | | |
| **SOC** | $\chi^2(2)=6.6$ | 0.04 | 0.03 | 0.7 | 0.2 |
| SOC (0-30 cm) | $\chi^2(2)=14.5$ | 0.0001 | 0.0001 | 0.5 | 0.009 |
| SOC (>30 cm) | $\chi^2(2)=3.0$ | 0.2 | - | | |
| **Cumul. CStock** | | | | | |
| CStock (0-30 cm) | $\chi^2(2)=5.4$ | 0.07 | 0.06 | 0.4 | 0.6 |
| CStock (0-100 cm) | $\chi^2(2)=5.4$ | 0.06 | 0.2 | 0.06 | 0.8 |
| **Bulk density** | $\chi^2(2)=3.4$ | 0.2 | - | | |
| Bulk density (0-30 cm) | $\chi^2(2)=10.3$ | 0.06 | 0.02 | 0.09 | 0.4 |
| Bulk density (>30 cm) | $\chi^2(2)=2.0$ | 0.4 | - | | |
| **C/N ratio** | $\chi^2(2)=5.9$ | 0.05 | 0.06 | 0.9 | 0.1 |
| C/N ratio (0-30 cm) | $\chi^2(2)=15.0$ | 0.0005 | 0.0002 | 0.8 | 0.003 |
| C/N ratio (>30 cm) | $\chi^2(2)=2.2$ | 0.3 | - | | |
| **H/C ratio** | $\chi^2(2)=6.7$ | 0.04 | 0.5 | 0.4 | 0.02 |
| H/C ratio (0-30 cm) | $\chi^2(2)=6.3$ | 0.04 | 0.6 | 0.4 | 0.03 |
| H/C ratio (>30 cm) | $\chi^2(2)=3.5$ | 0.2 | - | | |
| **O/C ratio** | $\chi^2(2)=14.0$ | 0.0009 | 0.006 | 0.7 | 0.0002 |
| O/C ratio (0-30 cm) | $\chi^2(2)=10.5$ | 0.005 | 0.06 | 0.7 | 0.003 |
| O/C ratio (>30 cm) | $\chi^2(2)=8.5$ | 0.014 | 0.008 | 1.0 | 0.003 |
| **$CO_2$ 10°C** | $\chi^2(2)=0.8$ | 0.7 | - | | |
| $CO_2$ 10°C (0-30 cm) | $\chi^2(2)=2.7$ | 0.3 | - | | |
| $CO_2$ 10°C (30-60 cm) | $\chi^2(2)=4.9$ | 0.09 | - | | |
| $CO_2$ 10°C (>60 cm) | $\chi^2(2)=1.8$ | 0.4 | - | | |
| **$CO_2$ 20°C** | $\chi^2(2)=1.4$ | 0.5 | - | | |
| $CO_2$ 20°C (0-30 cm) | $\chi^2(2)=6.5$ | 0.04 | 0.03 | 0.2 | 0.7 |
| $CO_2$ 20°C (30-60 cm) | $\chi^2(2)=1.7$ | 0.4 | - | | |
| $CO_2$ 20°C (>60 cm) | $\chi^2(2)=1.2$ | 0.5 | - | | |
| **Q10** | $\chi^2(2)=3.5$ | 0.2 | - | | |
| Q10 (0-30 cm) | $\chi^2(2)=0.4$ | 0.8 | - | | |
| Q10 (30-60 cm) | $\chi^2(2)=1.1$ | 0.6 | - | | |
| Q10 (> 30 cm) | $\chi^2(2)=1.5$ | 0.5 | - | | |

[1] P value of ANOVA comparing linear mixed models with and without the factor "land-use type".
[2] P value emitted using least square means between "land-use types"





**Table 3**

Tab. 3: Results of the depth influence analysis displayed for Q10 values , $CO_2$ emissions at 10 and 20°C, SOC contents, bulk densities, C/N ratios, H/C ratios and O/C ratios. Ea values acted showed similar significances as Q10 values.

| Attributes | Depth interaction | | P-values between specific depth classes | | |
|---|---|---|---|---|---|
| | $\chi^2$ Values | P-value | 0-30 vs 30-60 | 0-30 vs >60 | 30-60 vs >60 |
| **Q10 Values** | $\chi^2(2)=46.2$ | $9.56 \times 10^{-11}$ | 0.05 | <0.0001 | <0.0001 |
| Q10 Cropland | $\chi^2(2)=16.1$ | 0.0003 | 0.8 | 0.0002 | 0.002 |
| Q10 Forest | $\chi^2(2)=5.2$ | 0.08 | - | | |
| Q10 Grassland | $\chi^2(2)=29.5$ | $3.9 \times 10^{-7}$ | 0.06 | <0.0001 | 0.009 |
| **$CO_2$ emiss.(10°C)** | $\chi^2(2)=6.1$ | <0.05 | 0.03 | 0.7 | 0.2 |
| Cropland (10°C) | $\chi^2(2)=1.5$ | 0.5 | - | | |
| Forest (10°C) | $\chi^2(2)=17.3$ | 0.0001 | 0.0001 | 0.01 | 0.5 |
| Grassland (10°C) | $\chi^2(2)=7.9$ | 0.02 | 0.01 | 0.3 | 0.5 |
| **$CO_2$ emiss.(20°C)** | $\chi^2(2)=0.9$ | 0.6 | - | | |
| Cropland (20°C) | $\chi^2(2)=8.4$ | 0.015 | 0.02 | 1.0 | 0.09 |
| Forest (20°C) | $\chi^2(2)=13.2$ | 0.0001 | 0.0007 | <0.05 | 0.6 |
| Grassland (20°C) | $\chi^2(2)=3.5$ | 0.17 | - | | |
| **pH** | $\chi^2(2)=6.0$ | <0.05 | <0.08 | 0.15 | 1.5 |
| Cropland | $\chi^2(2)=19.4$ | $6.2 \times 10^{-5}$ | <0.02 | <0.0001 | 0.09 |
| Forest | $\chi^2(2)=36.8$ | $1 \times 10^{-8}$ | 0.004 | <0.0001 | 0.001 |
| Grassland | $\chi^2(2)=27.4$ | $1.1 \times 10^{-6}$ | 0.0001 | <0.0001 | 0.9 |
| **SOC** | $\chi^2(2)=321.0$ | $<2.2 \times 10^{-16}$ | <0.0001 | <0.0001 | 0.001 |
| Cropland | $\chi^2(2)=353.4$ | $<2.2 \times 10^{-16}$ | <0.0001 | <0.0001 | <0.0001 |
| Forest | $\chi^2(2)=13.8$ | <0.001 | 0.6 | 0.03 | 0.008 |
| Grassland | $\chi^2(2)=133.8$ | $<2.2 \times 10^{-16}$ | <0.0001 | <0.0001 | 0.6 |
| **Bulk Density** | $\chi^2(2)=254.1$ | $<2.2 \times 10^{-16}$ | <0.0001 | <0.0001 | <0.05 |
| Cropland | $\chi^2(2)=312.6$ | $<2.2 \times 10^{-16}$ | <0.0001 | <0.0001 | <0.0001 |
| Forest | $\chi^2(2)=31.6$ | $1.4 \times 10^{-7}$ | 0.7 | <0.0001 | <0.0001 |
| Grassland | $\chi^2(2)=121.6$ | $<2.2 \times 10^{-16}$ | <0.0001 | <0.0001 | 0.4 |
| **C/N ratio** | $\chi^2(2)=474.1$ | $<2.2 \times 10^{-16}$ | <0.0001 | <0.001 | <0.0001 |
| Cropland | $\chi^2(2)=431.4$ | $<2.2 \times 10^{-16}$ | <0.0001 | <0.001 | <0.0001 |
| Forest | $\chi^2(2)=35.3$ | $2.1 \times 10^{-8}$ | 0.4 | <0.001 | <0.0001 |
| Grassland | $\chi^2(2)=199.4$ | $<2.2 \times 10^{-16}$ | <0.0001 | <0.001 | <0.0001 |
| **H/C ratio** | $\chi^2(2)=66.6$ | $3.4 \times 10^{-15}$ | <0.0001 | <0.0001 | 1.0 |
| Cropland | $\chi^2(2)=46.7$ | $7.3 \times 10^{-11}$ | <0.0001 | <0.0001 | 0.32 |
| Forest | $\chi^2(2)=1.9$ | 0.38 | | | |
| Grassland | $\chi^2(2)=32.4$ | $9.4 \times 10^{-8}$ | <0.0001 | <0.0001 | 0.84 |
| **O/C ratio** | $\chi^2(2)=30.1$ | $2.9 \times 10^{-7}$ | <0.0001 | 0.004 | 0.5 |
| Cropland | $\chi^2(2)=12.7$ | 0.002 | 0.03 | 0.0009 | 0.32 |
| Forest | $\chi^2(2)=4.8$ | 0.09 | | | |
| Grassland | $\chi^2(2)=19.3$ | $6.6 \times 10^{-5}$ | <0.0001 | <0.05 | 0.24 |





**Table 4**

Tab. 4: Coefficients of determination for $CO_2$ emissions measured at 20°C and different soil attributes as explanatory variables (profile depth, SOC content, bulk density, C/N, O/C and H/C ratio).

| Attribute | 0-30 cm | | | | 30-100 cm | | | |
|---|---|---|---|---|---|---|---|---|
| $CO_2$ at 20°C | Intercept | cor | P value | $R^2$ | Intercept | cor | P value | $R^2$ |
| Depth (cm) | $<2.0 \times 10^{-16}$ | -0.23 | 0.01 | 0.05 | 0.0001 | 0.11 | 0.2 | 0.01 |
| SOC (g kg$^{-1}$) | $8.06 \times 10^{-6}$ | 0.31 | 0.001 | 0.09 | $7.6 \times 10^{-6}$ | -0.01 | 0.9 | $4.8 \times 10^{-5}$ |
| Bulk density (g kg$^{-1}$) | $<2.0 \times 10^{-16}$ | -0.27 | 0.003 | 0.07 | $<2 \times 10^{-16}$ | -0.19 | 0.02 | 0.04 |
| C/N ratio | 0.002 | 0.28 | 0.002 | 0.08 | $3.5 \times 10^{-6}$ | -0.1 | 0.5 | 0.004 |
| O/C ratio | $1.33 \times 10^{-6}$ | -0.02 | 0.9 | 0.0002 | $2.1 \times 10^{-5}$ | 0.06 | 0.4 | 0.01 |
| H/C ratio | 0.046 | 0.03 | 0.8 | 0.0005 | 0.292 | 0.07 | 0.4 | 0.01 |
| pH | $2.3 \times 10^{-8}$ | -0.29 | 0.001 | 0.09 | $2.3 \times 10^{-8}$ | -0.25 | 0.003 | 0.06 |
| $CO_2$ at 10°C | | | | | | | | |
| Depth (cm) | $1.7 \times 10^{-13}$ | 0.19 | 0.2 | 0.02 | 0.0002 | 0.10 | 0.2 | 0.02 |
| SOC (g kg$^{-1}$) | $1.7 \times 10^{-5}$ | -0.13 | 0.04 | 0.03 | $4.8 \times 10^{-16}$ | -0.24 | 0.002 | 0.06 |
| Bulk density (g kg$^{-1}$) | $<2.0 \times 10^{-16}$ | -0.22 | 0.02 | 0.04 | $<2.0 \times 10^{-16}$ | -0.01 | 0.9 | $9.1 \times 10^{-5}$ |
| C/N ratio | 0.0007 | 0.31 | 0.0007 | 0.09 | $1.05 \times 10^{-5}$ | -0.03 | 0.7 | 0.0008 |
| O/C ratio | $3.8 \times 10-5$ | 0.01 | 1.0 | $3.4 \times 10^{-5}$ | $3.4 \times 10^{-5}$ | 0.10 | 0.2 | 0.01 |
| H/C ratio | 0.5 | 0.14 | 0.1 | 0.2 | 0.3 | 0.24 | 0.002 | 0.06 |
| pH | $6.6 \times 10^{-5}$ | -0.14 | 0.1 | 0.02 | 0.0001 | -0.09 | 0.3 | 0.007 |
| Q10 values | | | | | | | | |
| Depth (cm) | $<2.0 \times 10^{-16}$ | 0.18 | 0.12 | 0.01 | 0.0001 | -0.30 | 0.0001 | 0.08 |
| SOC (g kg$^{-1}$) | $<2.0 \times 10^{-16}$ | -0.16 | 0.03 | 0.03 | $<2.0 \times 10^{-16}$ | 0.12 | 0.06 | 0.02 |
| Bulk density (g kg$^{-1}$) | $<2.0 \times 10^{-16}$ | -0.02 | 0.6 | -0.006 | $<2.0 \times 10^{-16}$ | -0.02 | 0.4 | -0.002 |
| C/N ratio | $8.1 \times 10^{-16}$ | -0.10 | 0.15 | 0.01 | $1.2 \times 10^{-15}$ | -0.08 | 0.4 | -0.002 |
| O/C ratio | $5.3 \times 10^{-15}$ | 0.20 | 0.07 | 0.02 | $<2.0 \times 10^{-16}$ | -0.13 | 0.3 | 0.0002 |
| H/C ratio | 0.0001 | 0.06 | 0.2 | 0.004 | $2.9 \times 10^{-8}$ | -0.13 | 0.06 | 0.02 |
| pH | $1.02 \times 10^{-9}$ | -0.03 | 0.8 | 0.0006 | $2.7 \times 10^{-8}$ | -0.02 | 0.8 | 0.0003 |





1  **Table 5**

2  Tab. 5: Incubation studies with organic and mineral soils at different moisture levels, soil depths and temperatures. If moisture level says moist, samples were incubated directly after
3  being retrieved from the field, while saturated samples were incubated under wetter, i.e. anaerobic conditions.

| | Soil | Region | Moisture level | °C | days | C/N | depths (cm) | $CO_2$ emissions (mg $CO_2$-C g$^{-1}$ SOC d$^{-1}$) | $Q_{10}$ | $E_a$ (kJ mol$^{-1}$) |
|---|---|---|---|---|---|---|---|---|---|---|
| This study | drained Fens | Switzerland | -10 kPa | 10<br>20 | 416 | 17.7 | 5-150 | 0.078<br>0.18 | 2.57 | 69.5 |
| (Chapman & Thurlow, 1998) | Drained/undrained Bogs | UK (Scotland) | Moist | 10<br>20 | unknown | - | 0-20 | 0.051*<br>0.030* | 3.2 | 80.0 |
| (Grover & Baldock, 2012) | Bog | Australia | Moist | 20 | 38 | 15-25 | 5-110 | 0.13-0.78 | - | |
| (Hahn-Schöfl et al., 2011) | Fen | Germany | Saturated | 20 | 346 | 15.3 | 5-110 | 0.013 | - | |
| (Hardie et al., 2011) | Bog | UK | Dryer | 5<br>10<br>15 | 6 | 30 | 0-30 | 0.027<br>0.049<br>0.093 | 3.66 | 86.4 |
| (Hartley & Ineson, 2008) | Mineral soil | UK | Dryer | 10<br>20 | 124 | | unclear | 0.046<br>0.074 | 3.25 | 81.3 |
| (Hilasvuori et al., 2013) | Bog | Finland | Moist | 10<br>20 | short | 83 | 0-44 | 0.016*<br>0.061* | 2 | 22.7 |
| (Hogg et al., 1992) | Fen | Canada | Similar | 8<br>16<br>24 | 120 | 40.6 | 5-40 | 0.083<br>0.282<br>0.381 | 1.9 -2.2 | 62.0 |
| (Karhu et al., 2014) | Organic soil | UK (Scotland) | Similar | 11.4 | 174 | 28.6 | 0-10 | 0.065 | - | |
| | Organic soil | UK (Scotland) | | 7.6 | 174 | 36.5 | 0-10 | 0.105 | - | |
| | Organic soil | UK | | 13.3 | 174 | 18.7 | 0-10 | 0.201 | - | |
| | Mineral soil | UK | | 11.4 | 174 | 13.3 | 0-10 | 0.101 | - | |
| | Mineral soil | Spain | | 21.5 | 174 | 14.3 | 0-10 | 0.293 | - | |
| | Mineral soil | Spain | | 19 | 174 | 13.0 | 0-10 | 0.345 | - | |
| | Mineral soil | Spain | | 19 | 174 | 18.6 | 0-10 | 0.448 | - | |
| | Mineral soil | Italy | | 18.4 | 174 | 13.2 | 0-10 | 0.129 | - | |
| (Karhu et al., 2010) | Mineral soil | Finland | Similar | 8-25 | 540 | - | 0-30 | - | 3.0 | 45.0 |
| (Koch et al., 2007) | Organic soil | Austria | Moist | 0-30 | 25 | 21.6 | 0-5 | - | 2.0 | 31.9 |
| (Leifeld & Fuhrer, 2005) | Mineral soil | Switzerland | Similar | 25 | 707 | 7.85 | 5-35 | 0.12 | 4.6 | 110.8 |
| (Neff & Hooper, 2002) | Organic soil | USA (Alaska) | unclear | 10<br>30 | 352 | 34.6 | 0-10 | 0.32<br>0.75 | 1.9 | 22.9 |
| (Plante et al., 2010) | Mineral soil | USA | Similar | 15 | 56 | | 0-20 | 0.28 | 1.36  -  1.79 | 31.7 |
| (Reiche et al., 2010) | Fen | Germany | Saturated | 15 | 31 | 30.1 | 0-40 | 0.0022 | - | |
| (Reichstein et al., 2000) | Mineral soil | Switzerland | Similar | 5<br>15<br>25 | 104 | 30.3 | | 0.05<br>0.14<br>0.22 | 2.5-2.7 | 65.9 |
| (Scanlon & Moore, 2000) | Fen | Canada | Moist | 4<br>14 | 12 | 43 | 5-45 | 0.227<br>0.109 | 2.0 | 45.8 |
| (Wang et al., 2010) | Organic soil | China | Similar | 5-20 | 40 | 28.5 | 10-30 | 0.31 | 2.2 | 53.3 |
| (Wickland & Neff, 2008) | Organic soil | Canada | Similar | 10<br>20 | 57 | 24.7 | 2-30 | 0.35<br>0.79 | 1.7 | 36.6 |
| (Yavitt et al., 2000) | Bog | Canada | Moist | 12-22 | 2 | | 0-54 | - | 1.4 | 32.5 |

* Study authors are not specific about the SOC content of peat, therefore we assumed it to be 440 gkg$^{-1}$, according to the results of undisturbed bog peat from the same site in Laine et al. (2004).