# Peer review of "Peat decomposability in managed organic soils in relation to land-use, organic matter composition and temperature"

_Biogeosciences, 2017_

## Referee Comment (RC1) · Anonymous Referee #1 · 25 Jul 2017

The manuscript of Bader et al. presents information of peat decomposability in managed organic soils in relation to land-use, organic matter composition and temperature. The paper is well structured and written in fluent language. However, I have few minor and mainly technical comments and suggestions.

Major comments:

1) In materials and methods, more information of the sites, sampling design and samples treatments is needed. The requested new information is listed in the minor comments.

2) Site codes must be uniform throughout the manuscript!

3) Should add more literature about the organic matter properties and how it regulates/does not regulate decomposition processes.

4) This study focuses mainly on the effects of drainage on decomposition and SOM characteristics, yet the differences in management (i.e. machinery tilling, fertilization) between sites will likely affect to peat decomposability and decomposition as well. Now these management related differences are mentioned first time in the discussion section, but they should be mentioned already at the site description in the materials and methods.

Especially, the current vegetation type (forest, grass, crops) undoubtedly has influence on the current peat properties due to differences in litter quality and quantity. Additionally, in cropland the fertilization will affect to nutrient availability, and thus likely influences on decomposition. At least the variation in litter input should be discussed more detailed.

Minor changes:

Line 33: It is not necessary to Term "peatland" means organic soils, thus "containing organic soils" should be removed.

Line 34: The word "destroyed" should be replaced with something more neutral.

Line 34: Modify to "Drainage aerates the soil. . ."

Line 101: 20 % of carbon released/accumulated in/from organic soils under agriculture

Line 120: I could not find information of the size of the peatlands from the text or the tables. This could be added to Table 1. Additionally, no information of the current water table depths, which expresses the intensity of the drainage and determines the depth of aerobic layer in peat profile, can be found from the manuscript. Please, add this information.

Line 120: Where all the sites originally tree covered?

Line 125: Uniform site markings. Here you use SKF, KF and elsewhere you use SK_FL and K_FL. Double check this throughout manuscript. Additionally, the sites should be organized alphabetically in Table 1 and Supplementary Table1.

Line 127-130: How were the sampling spots chosen?

Line 128: From what site were 8 cores taken and why?

Line 128-130: Rephrase the sentences. Clarify also in which site and why you cored deeper than at other sites on this particular site?

Line 130: What is the volume of the samples?

Line 134: I could not find information on which cores you found mineral sediments. This information could be added for example to Table 1 or Supplementary table.

Line 142: Specify from which samples you conducted the analyses. Did you analyze subsamples from all intersections of each core?

Line 154: How did you select the soil segments? Were they always from the same depths – if not, why weren't they?

Line 155: Did you have replicates of the samples from the same site and depth in different temperatures or did you use field replicates?

Line 156: Again, uniform the site markings. Here you use MCL and in the tables and figures something else.

Line 170: Is it really appropriate to remove the negative values?

Line 179: Modify the citing or remove the brackets around the references:" used by e.g. Hogg et al. (1992), Scanlon & Moore (2000), Wang et al. (2010) . . ."

Line 210: Where was the land-use effect significant? In the topmost 30 cm?

Line 215: Do not start the sentence with words like also.

Line 231: Replace word lost with the word emitted

Line 232 and Line 236: Again, the site codes are different than in the Table 1 and in the supplementary table.

Line 236: Double check the site comparisons in the brackets.

Lines 257, 259, 264, 298 and elsewhere in discussion: Consider depersonalizing when you talk about the soil profiles and samples. I.e. In the line 257 the part " our soils profiles were close..." could be modified to "the soil profiles were close to values that are typical for undisturbed peat".

Line 260: Are the managed sites also tilled and fertilized? This also triggers decomposition. After a long time passes from drainage, and the decomposition processes improved by enhanced oxygen conditions stabilizes, tilling and fertilizing may become even more important factors controlling the decomposition than the lowered water table.

Line 272: Add the management (tilling, potential fertilizing, harvesting...) information of the sites to table 1.

Line 292-295: You could discuss also of the effects of management. According to the earlier parts of the discussion section, at least the cropland has been fertilized and tilled with machines. This has likely affected to amount of nutrient availability for decomposers, and thus potentially triggered the decomposition processes.

Line 298: Depersonalize. For examples:" The samples lost, on average... ".

Line 298: incubation over one year

Line 314: Please, add the information of the drainage depths to the Table 1. This is important information as the whole study is about drained peatlands.

Line 327: Again, the site codes are different than in the Table 1. Also mention clearly, that these are the same sites as in this study.

Line 343: below the 60 cm depth

Figure 2: You could set the sampling depths (0-30 cm and <30 cm) as headings on the top of the panels.

Figure 3: This figure is extremely hard to follow. Please reorganize the sites on the left first by land-use type and then alphabetically. Make sure the site coding is uniform throughout the manuscript.

Figure 4: You could bold the blue symbols indicating organic soils. That would make the figure readable even as black and white prints.

Table 1. Organize the sites alphabetically and make sure the coding is uniform throughout the manuscript. Is it necessary to write the abbreviation of the land use type also in the first column after the site abbreviation? Please add the water table depth information either here or somewhere, but just add it. Add information about other land management history beside the drainage (tillage, harvesting, fertilization) and estimations of the intensity. Do you have any estimations of the size of the sites? That would be interesting additional information.

Table 2. I could not find where the footnotes belonged to.

Table 5. Please modify the references in the first column as following: Chapman & Thurlow (1998).

Table 5. In the column expressing "Moisture level" on few rows (e.g. Hogg et al. 1992) is text "Similar". How is this defined and what does it mean? Table S1./ Supplementary table. Please organize the sites alphabetically or specify clearly why it is in order that not seem to make any sense for a new reader.

---

## Referee Comment (RC2) · Anonymous Referee #2 · 9 Aug 2017

This manuscript describes the decomposability of organic soils (previously peat) from 21 sites in Switzerland that are now managed as cropland, grassland, and forest. The study looks at CO2 emission rates from laboratory incubations and possible correlations to various soil characteristics and temperature. In general, the study is well performed and provides a lot of details, however I have same major comments:

1) I find the term peat and peat decomposability quite misleading. While it might have been peat at some point it is not peat anymore and drainage must have occurred a long time ago (according to Table 1 sometimes 150 years ago). The authors do not address the history of the peat in the sampled locations adequately or give the reader

the proper background in regard to changes from peat to grassland, cropland, and forest. Table 1 has some information on the drainage history of the sites but it is not mentioned anywhere in the text. The introductory part about peatlands becoming cropland, grassland, or forest (p. 2) is too general and does not specifically address the sites. I also find it confusing how peatland and organic soil is used interchangeable (so it seems to me) in the manuscript, not every organic soil is a peatland. I think referring to organic soils (and it needs to be clearly defined at the beginning of the introduction what organic soils are, which is not there right now) throughout the text would be more appropriate.

2) The statistical analyses are not well enough explained and from what I understand not the appropriate analysis is performed. Why not perform a full linear mixed-effects model that includes all soil characteristics as fixed effects (land-use type, pH, bulk density, C/N etc.) in the same model while including depth and sampling location as random variables? Then, the model could be reduced step by step and each sub-model gets compared to the full model and by using the smallest AIC as the model selection criterion, it will be possible to identify the variable that has the strongest influence on CO2 release. Of course the variables included need to be tested for collinearity (e.g. total carbon and C/N most likely correlate and only one variable can be included). Given the lack of detail for the statistical analysis I could not make much sense of all the tables but in general, I find it very commendable if so much detail is provided in tables.

3) Overall, I am missing a story line and focus that brings the message across in an easily understandable way. The result section reads like a listing of findings and there is no result that gets highlighted or seems particularly memorable. I am also missing a link to the global scale, which I was expecting since the authors start out the introduction with the importance of organic soils globally.

Smaller comments:

4) What type of cropland is represented in this study? It never says which crop it is and I wonder if wheat compared to corn or other crops might be different

5) I do not understand the usefulness for Figure 4

6) P. 5, l. 158, I assume thoroughly mixed means homogenized? If so, bulk density as a variable loses its meaning completely

7) Fig. 1. The y-axis for all panels is depth (cm) but it is not written anywhere

8) Fig. 1, the symbols are so small, I think using larger symbols and different shades of black and white would really help the readability of this graph

9) Fig. 3, I think it would be much more useful to keep the same order for each sampling location for the upper and deeper soil layer, maybe keep the left panel (upper soil layer) as is and only adjust the right panel.

10) What about CH4 from any of these sites? If hydrology matters as much as the authors write then I would expect to read something about CH4 release

---

## Author Comment (AC1) · 17 Aug 2017

Answer to the reviewer comments on the article "Peat decomposability in managed organic soils in relation to land-use, organic matter composition and temperature" by Cédric Bader et al.

We thank the two referees for their profound study of our manuscript, their helpful suggestions and their positive perception of our work. We give here information on how we plan to revise our article and to proceed. R (referee comment), A (author reply)

Referee #1

Major comments:

R: In materials and methods, more information of the sites, sampling design and samples treat-ments is needed. The requested new information is listed in the minor comments.

A: Referee #1 suggests that the size of each peatland is added to Table 1. This information will be included upon revision. Further, the referee states that there is no information on water table or drainage depths. Unfortunately we do not have this information. Yet, we know that the extensively managed sites are drained only at the surface. This information will be added to Table 1. We will further add a supplementary table which describes the soil profile down to a "permanent" mineral layer.

R: Site codes must be uniform throughout the manuscript!

A: We thank the reviewer for careful reading and will change our manuscript accordingly.

R: Should add more literature about the organic matter properties and how it regulates/does not regulate decomposition processes.

A: We will add some more literature on this topic.

R: This study focuses mainly on the effects of drainage on decomposition and SOM characteristics, yet the differences in management (i.e. machinery tilling, fertilization) between sites will likely affect peat decomposability and decomposition as well. Now these management related differences are mentioned first time in the discussion section, but they should be mentioned already at the site description in the materials and methods. Especially, the current vegetation type (forest, grass, crops) undoubtedly has influence on the current peat properties due to differences in litter quality and quantity. Additionally, in cropland the fertilization will affect to nutrient availability, and thus likely influences on decomposition. At least the variation in litter input should be discussed

more detailed.

A: We will include a description of management practices in the materials and methods section. Regarding the effect of vegetation type on peat composition we already argue, that the litter inputs likely have caused a different chemical composition of organic matter in forest as compared to agricultural topsoil (H/C and O/C ratios). In our discussion of these effects we however stress, that they are no major driver for peat decomposability. Yet, we do not know the exact fractions of litter derived vs. peat derived SOM. Therefore, we find a more detailed discussion on possible effects of litter input quality on SOM decomposability speculative.

Requested minor changes:

R: Referee #1 calls for a number of minor changes and corrections that are not listed here separately.

A: We concur with almost all minor comments and will be happy to improve our manuscript accordingly, also by including missing information. One minor remark refers to our decision to omit negative $CO_2$ values (0.45 % of all measurements). We believe that this data treatment is plausible. The omitted values were mostly strongly negative and occurred as single events during very short time spans, suggesting that they represent electronic artifacts rather than being product of a biological process.

Referee #2

Major comments:

R: I find the term peat and peat decomposability quite misleading. While it might have been peat at some point it is not peat anymore and drainage must have occurred a long time ago (according to Table 1 sometimes 150 years ago). The authors do not address the history of the peat in the sam-pled locations adequately or give the reader the proper background in regard to changes from peat to grassland, cropland, and forest. Table 1 has some information on the drainage history of the sites but it is

not mentioned anywhere in the text. The introductory part about peatlands becoming cropland, grassland, or forest (p. 2) is too general and does not specifically address the sites. I also find it confusing how peatland and organic soil is used interchangeable (so it seems to me) in the manuscript, not every organic soil is a peatland. I think referring to organic soils (and it needs to be clearly defined at the beginning of the introduction what organic soils are, which is not there right now) throughout the text would be more appropriate.

A: All of our soils are Histosols according to WRB, and organic matter accumulating in histic horizons is termed peat. This also holds true when the peatland is drained, i.e., degrading. In an earlier study, conducted at one of our sites (Bader et al. 2017, cited in MS), we showed that a major fraction of SOM still originates from old peat. Therefore, we find the terms peat and peat decomposability appropriate. We however agree with the reviewer that using the terms peatland and organic soils interchangeably can be misleading. The sites used for this study are, from an ecological viewpoint, no intact peatlands as they do not accumulate new peat anymore. Therefore we will consistently move to the term organic soils. While we can provide some information whether the peat was derived from a bog or a fen, being more specific on the land-use and drainage history as currently given in Table 1 is very difficult. We will however, also in response to the first reviewer, provide an estimate on the time sites are managed with the current land-use.

R: The statistical analyses are not well enough explained and from what I understand not the appropriate analysis is performed. Why not perform a full linear mixed-effects model that includes all soil characteristics as fixed effects (land-use type, pH, bulk density, C/N etc.) in the same model while including depth and sampling location as random variables? Then, the model could be reduced step by step and each submodel gets compared to the full model and by using the smallest AIC as the model selection criterion, it will be possible to identify the variable that has the strongest influence on $CO_2$ release. Of course the variables included need to be tested for collinearity (e.g.

total carbon and C/N most likely correlate and only one variable can be included). Given the lack of detail for the statistical analysis I could not make much sense of all the tables but in general, I find it very commendable if so much detail is provided in tables.

A: We thank referee #2 for these suggestions. We will perform a full linear mixed-effects model using fixed effects such as SOC, nitrogen and oxygen content as well as the bulk density. Further we will implement the land-use type as fixed effect. We will use the data on the single elements rather than the ratios in order to avoid collinearity. We will not use pH values because we do have only pH measurements for the different profiles but not for each single sample used for incubation.

R: Overall, I am missing a story line and focus that brings the message across in an easily understandable way. The result section reads like a listing of findings and there is no result that gets high-lighted or seems particularly memorable. I am also missing a link to the global scale, which I was expecting since the authors start out the introduction with the importance of organic soils globally.

A: What we do highlight, and this will be stressed even more upon revision, is that none of the parameters we measured was able to explain a substantial proportion of the variability in $CO_2$ release rates. We also write in the abstract: 'This, in turn, indicates a relative accumulation of recalcitrant peat in topsoils. Hence, our data suggest that after exposure of subsoil peat in the future, carbon loss from agriculturally managed organic soils will be similar considering warmer climate conditions.' We carefully considered whether our data allow such a conclusion and believe, this is a fair range of interpretation. These sentences do not support the referee's viewpoint that no result seems particularly memorable. We, of course, introduce to the topic by starting at the global scale at which the relevance of peatlands and organic soils is well acknowledged. This gives the motivation for our study. According to our knowledge, there has not been a previous work that included such a wide array of samples in the analysis of peat decomposability and its drivers.

Requested minor changes:

We thank the referee for his/her careful reading and will address all minor points when revising our MS. One point to be highlighted at this stage is the referee's question on the usefulness of our Figure 4. That figure, which includes comparison to $CO_2$ release rates from other studies including both, mineral and organic soils, is, from our viewpoint, particularly relevant. It puts our results into a broader context. We write: 'Therefore, the pools size of labile carbon, indicated by the decomposition rates, seem not to differ between these soil classes. This comparison suggests that accumulation of recent, labile plant materials that presumably account for most of the evolved $CO_2$ is not systematically different between mineral and organic soils.' The comparison of organic matter decomposability in mineral and organic soils is highly relevant because, at present, the discussion on mineral soil carbon sequestration is tightly motivated by various so-called stabilization mechanisms. Our data indicate that OM decomposability seems not to differ systematically between these major soil groups and will hopefully inspire future discussion on the mechanisms of action.

---

## Author Response (AR1)

**Answer to the reviewer comments on the article** *"Peat decomposability in managed organic soils in relation to land-use, organic matter composition and temperature"* **by Cédric Bader et al.**

We thank the two referees for their profound study of our manuscript and their helpful suggestions. Below we provide a point-by-point response to all comments (R – reviewer comment, A – authors' response). We hope that with these revisions our MS will be suitable for publication in Biogeosciences.

**Referee #1**

**Major comments:**

R: In materials and methods, more information of the sites, sampling design and samples treatments is needed. The requested new information is listed in the minor comments.

R: Referee #1 suggested that the size of each peatland is added to table 1.

A: We added the area of each peatland to table 1.

R: Further, referee # 1 stated that there is no information on water table or drainage depths.

A: This information is unfortunately not available in detail as it requires long-term records. Yet, we know that the extensively managed sites are drained only on the surface and provide an assignment to drainage class shallow (< 0.5 m) and deep (> 0.5m) for all sites which refers to the situation in the field at sampling.

R: Site codes must be uniform throughout the manuscript!

A: We thank reviewer 1 for this remark and unified site codes accordingly.

R: Should add more literature about the organic matter properties and how it regulates/does not regulate decomposition processes.

A: We addressed this issue already extensively in the text but added two more references and one sentence in the introduction to this topic.

R: This study focuses mainly on the effects of drainage on decomposition and SOM characteristics, yet the differences in management (i.e. machinery tilling, fertilization) between sites will likely affect to peat decomposability and decomposition as well. Now these management related differences are mentioned first time in the discussion section, but they should be mentioned already at the site description in the materials and methods.

A: We added information on management to the methods section.

R: Especially, the current vegetation type (forest, grass, crops) undoubtedly has influence on the current peat properties due to differences in litter quality and quantity. Additionally, in cropland the

fertilization will affect to nutrient availability, and thus likely influences on decomposition. At least the variation in litter input should be discussed more detailed.

A: We are aware of possible differences in litter input among the land use types and argue, that litter inputs might have altered the chemical composition of topsoils OM (H/C and O/C ratios) differently for the three land use types. This is already discussed in section 3.1. Chemical analysis of SOM composition shows that there was no clear relationship to organic matter decomposability. Yet, we do not know the exact fractions of litter derived SOM or peat derived SOM. This is mentioned as one possible factor for variations in $CO_2$ release in our manuscript, also with reference to recent work. We find a more detailed discussion of the litter inputs to be too speculative and remain to say that those SOM parameters that we measured, and that differed between layers and land use types, are no strong explanatory variables for $CO_2$ production.

Minor changes:

We thank Referee #1 for the corrections suggested for our manuscript. We do concur with most of the minor comments and improved the text accordingly. Below, responses are always placed behind the reviewer comment.

Line 33: It is not necessary to term "peatland" means organic soils, thus "containing organic soils" should be removed. Done

Line 34: The word "destroyed" should be replaced with something more neutral. Replaced by degraded. Done.

Line 34: Modify to "Drainage aerates the soil. . ." Done

Line 101: 20 % of carbon released/accumulated in/from organic soils under agriculture. Re-arranged.

Line 120: I could not find information of the size of the peatlands from the text or the tables. This could be added to table 1. Additionally, no information of the current water table depths, which expresses the intensity of the drainage and determines the depth of aerobic layer in peat profile, can be found from the manuscript. Please, add this information. Information on peatland size and drainage intensity has been added to table 1.

Line 120: Where all the sites originally tree covered? Site information was added to methods section.

Line 125: Uniform site markings. Here you use SKF, KF and elsewhere you use SK_FL and K_FL. Double check this throughout manuscript. Additionally, the sites should be organized alphabetically in table 1 and supplementary table1. Done.

Line 127-130: How were the sampling spots chosen? Explained now in M&M.

Line 128: From what site were 8 cores taken and why? Site G_GL because we used it for another study as well. We mentioned this in the discussion version of our MS for reasons of correctness although it is not relevant for the presented results. Upon revision, we omitted that information in order to not confuse the reader.

Line 128-130: Rephrase the sentences. Clarify also in which site and why you cored deeper than at other sites on this particular site? The deeper coring was done for the purpose of another study and the data of below 1 m depth are not included in the current MS. We changed the text in order to not confuse the reader.

Line 130: What is the volume of the samples? This depended on the type of the corer we used. In any case, the volume was known. We added information to that to the methods section.

Line 134: I could not find information on which cores you found mineral sediments.

This information could be added for example to Table 1 or Supplementary table. We added more information on how the identification of mineral layers was done. Because mineral layers occurred for most of the studied cores and because they were not included in the incubation, we refrain from listing them separately by name.

Line 142: Specify from which samples you conducted the analyses. Did you analyze subsamples from all intersections of each core? Information added to MS.

Line 154: How did you select the soil segments? Were they always from the same depths – if not, why weren't they? Most of the time they were. Depending on the occurrence of sediment layers or bigger roots we had to slightly adjust the specific depths (see suppl. table).

Line 155: Did you have replicates of the samples from the same site and depth in different temperatures or did you use field replicates? Sampled soil was always divided into two subsegments that were then incubated at the two temperatures.

Line 156: Again, uniform the site markings. Here you use MCL and in the tables and figures something else. Done

Line 170: Is it really appropriate to remove the negative values? We believe that this data treatment is plausible. First, the dark chamber measurements do not allow photosynthesis. The omitted values were mostly strongly negative and occurred as single events during very short time spans, suggesting that they represent electronic artifacts rather than being product of a biological process. Moreover, the share of negative values is only 0.45 % of all measurements).

Line 179: Modify the citing or remove the brackets around the references:" used by e.g. Hogg et al. (1992), Scanlon & Moore (2000), Wang et al. (2010). Referencing was adjusted according to journal requirements.

Line 210: Where was the land-use effect significant? In the topmost 30 cm? Yes the land-use effects was significant in the topmost 30 cm, indicated in text now.

Line 215: Do not start the sentence with words like also. Changed

Line 231: Replace word lost with the word emitted. Done

Line 232 and Line 236: Again, the site codes are different than in the Table 1 and in the supplementary table. Done

Line 236: Double check the site comparisons in the brackets. Checked

Lines 257, 259, 264, 298 and elsewhere in discussion: Consider depersonalizing when you talk about the soil profiles and samples. I.e. In the line 257 the part " our soils profiles were close. . ." could be modified to "the soil profiles were close to values that are typical for undisturbed peat". Good hint, we changed the wording at several places.

Line 260: Are the managed sites also tilled and fertilized? This also triggers decomposition. After a long time passes from drainage, and the decomposition processes improved by enhanced oxygen conditions stabilizes, tilling and fertilizing may become even more important factors controlling the decomposition than the lowered water table. We added information to site management to the M&M section.

Line 272: Add the management (tilling, potential fertilizing, harvesting. . .) information of the sites to table 1. Information was added to the site description, not to Table 1.

Line 292-295: You could discuss also of the effects of management. According to the earlier parts of the discussion section, at least the cropland has been fertilized and tilled with machines. This has likely affected to amount of nutrient availability for decomposers, and thus potentially triggered the decomposition processes. We agree, but the management information is not precise enough to distinguish specific management factors (fertilization, tillage, rotation) in their impact on peat decomposition. Particularly, management effects on peat decomposition last for decades, and management records of that lengths are not available. For example, tillage may have changed over the years. This is why we focused our analysis on the three land use types. They also comprise different management practices and we know that sites were managed under their particular land use for decades.

Line 298: Depersonalize. For examples:" The samples lost, on average. . . ". Done

Line 298: incubation over one year. Changed

Line 314: Please, add the information of the drainage depths to the Table 1. This is important information as the whole study is about drained peatlands. Information added

Line 327: Again, the site codes are different than in the Table 1. Also mention clearly, that these are the same sites as in this study. Done

Line 343: below the 60 cm depth. Done

Figure 2: You could set the sampling depths (0-30 cm and <30 cm) as headings on the top of the panels. Done

Figure 3: This figure is extremely hard to follow. Please reorganize the sites on the left first by land-use type and then alphabetically. Make sure the site coding is uniform throughout the manuscript. Done

Figure 4: You could bold the blue symbols indicating organic soils. That would make the figure readable even as black and white prints. Done

Table 1. Organize the sites alphabetically and make sure the coding is uniform throughout the manuscript. Is it necessary to write the abbreviation of the land use type also in the first column after the site abbreviation? Please add the water table depth information either here or somewhere, but just add it. Add information about other land management history beside the drainage (tillage, harvesting, fertilization) and estimations of the intensity. Do you have any estimations of the size of the sites? That would be interesting additional information. The order in Table 1 is now alphabetical. Column with abbreviations was modified. Estimated size of site and drainage class were added to table.

Table 2. I could not find where the footnotes belonged to. Done

Table 5. Please modify the references in the first column as following: Chapman & Thurlow (1998). Done

Table 5. In the column expressing "Moisture level" on few rows (e.g. Hogg et al. 1992) is text "Similar". How is this defined and what does it mean? Information added to table.

Table S1./ Supplementary table. Please organize the sites alphabetically or specify clearly why it is in order that not seem to make any sense for a new reader. All sites are identified by their code and we see no reason to put them in any particular new order. Different land use types at the same spot are arranged in consecutive order.

**Referee #2**

**Major comments:**

R. I find the term peat and peat decomposability quite misleading. While it might have been peat at some point it is not peat anymore and drainage must have occurred a long time ago (according to Table 1 sometimes 150 years ago). The authors do not address the history of the peat in the sampled locations adequately or give the reader the proper background in regard to changes from peat to grassland, cropland, and forest. Table 1 has some information on the drainage history of the sites but it is not mentioned anywhere in the text. The introductory part about peatlands becoming cropland, grassland, or forest (p. 2) is too general and does not specifically address the sites. I also find it confusing how peatland and organic soil is used interchangeable (so it seems to me) in the manuscript, not every organic soil is a peatland. I think referring to organic soils (and it needs to be clearly defined at the beginning of the introduction what organic soils are, which is not there right now) throughout the text would be more appropriate.

A. In consideration also of what we wrote as an online response to this comment, we screened our MS for consistence use of terms. We use the term 'peatland' when referring to the literature in cases the other authors use that term, but use 'organic soil' when referring to our own measurements and sites. We added site information to table 1: Table 1 also contains information on drainage history and, newly added, drainage intensity. In response to the first referee, we included information on typical management practices in the three land use types to the methods section. Unlike the referee states, we feel that we appropriately put our study into the context of land-use change. We did so in the beginning of the introduction, where we refer to the situation of land-use change on former peatlands in Europe and its impact on GHG emissions, and now also add a sentence with two references at the end of the introduction, indicating that our sites are representative for the situation of drained peatlands in Europe and Switzerland.

R. The statistical analyses are not well enough explained and from what I understand not the appropriate analysis is performed. Why not perform a full linear mixed-effects model that includes all soil characteristics as fixed effects (land-use type, pH, bulk density, C/N etc.) in the same model while including depth and sampling location as random variables? Then, the model could be reduced step by step and each sub-model gets compared to the full model and by using the smallest AIC as the model selection criterion, it will be possible to identify the variable that has the strongest influence on CO2 release. Of course the variables included need to be tested for collinearity (e.g. total carbon and C/N most likely correlate and only one variable can be included). Given the lack of detail for the statistical analysis I could not make much sense of all the tables but in general, I find it very commendable if so much detail is provided in tables.

A: We thank Referee #2 for these suggestions. In extension to what we replied already in the discussion section, we ran all the models since then using more site characteristics as fixed effects if the AIC score was smaller than before. For most of the tests the AIC score did not get smaller using more fixed effects. On few occasions the new models led to changes in the significances of the tests, which did not influence the broad picture reported by us. The modified results are now displayed in tables 2 and 3.

R. What type of cropland is represented in this study? It never says which crop it is and I wonder if wheat compared to corn or other crops might be different

A. We added information on crop rotations to the methods section. It must be recognized that crop rotations can be complex and change over time. Therefore, reporting the crop type at the occasion of sampling is not meaningful.

R. I do not understand the usefulness for Figure 4.

A. We responded to this point extensively already in our online reply.

R. P. 5, l. 158, I assume thoroughly mixed means homogenized? If so, bulk density as a variable loses its meaning completely.

A. Only the samples used for incubation were homogenized and we only homogenized material from the respective 5 cm segment. Bulk density results are part of Fig. 1 and reveal pronounced differences between land use types.

R. Fig. 1. The y-axis for all panels is depth (cm) but it is not written anywhere.

A. Thank you, information added to figure caption.

R. Fig. 1, the symbols are so small, I think using larger symbols and different shades of black and white would really help the readability of this graph.

A. Fig. 1 has been modified and symbols are enlarged.

R. Fig. 3, I think it would be much more useful to keep the same order for each sampling location for the upper and deeper soil layer, maybe keep the left panel (upper soil layer) as is and only adjust the right panel.

A. Sample order is now the same in both panels.

R. What about CH4 from any of these sites? If hydrology matters as much as the authors write then I would expect to read something about CH4 release.

A. Methane emission from peatlands typically drops substantially when drained and converted to agriculture or forestry. Anyhow, we present results from an incubation study under aerobic conditions and not from emissions in the field, where methane might play a role.